# Adaptation supports short-term memory in a visual change detection task

**Brian Hu** *, **Marina E. Garrett, Peter A. Groblewski, Douglas R. Ollerenshaw,
Jiaqi Shang, Kate Roll, Sahar Manavi, Christof Koch, Shawn R. Olsen,
Stefan Mihalas** *

Allen Institute for Brain Science, Seattle, Washington, United States of America

\* brian.hu@kitware.com (BH); stefanm@alleninstitute.org (SM)

pcbi.1009246

College de France, CNRS, FRANCE

**Data Availability Statement:** All the code for the
model and its analysis is available on github at:
https://github.com/AllenInstitute/STSPNet. The
physiology data is already available as it is part of a

## Abstract

The maintenance of short-term memories is critical for survival in a dynamically changing
world. Previous studies suggest that this memory can be stored in the form of persistent
neural activity or using a synaptic mechanism, such as with short-term plasticity. Here, we
compare the predictions of these two mechanisms to neural and behavioral measurements
in a visual change detection task. Mice were trained to respond to changes in a repeated
sequence of natural images while neural activity was recorded using two-photon calcium
imaging. We also trained two types of artificial neural networks on the same change detec-
tion task as the mice. Following fixed pre-processing using a pretrained convolutional neural
network, either a recurrent neural network (RNN) or a feedforward neural network with
short-term synaptic depression (STPNet) was trained to the same level of performance as
the mice. While both networks are able to learn the task, the STPNet model contains units
whose activity are more similar to the *in vivo* data and produces errors which are more simi-
lar to the mice. When images are omitted, an unexpected perturbation which was absent
during training, mice often do not respond to the omission but are more likely to respond to
the subsequent image. Unlike the RNN model, STPNet produces a similar pattern of behav-
ior. These results suggest that simple neural adaptation mechanisms may serve as an
important bottom-up memory signal in this task, which can be used by downstream areas in
the decision-making process.

## Author summary

Animals have to adapt to environments with rich dynamics and maintain multiple types
of memories. In this study, we focus on a visual change detection task in mice which
requires short-term memory. Learning which features need to be maintained in short-
term memory can be realized in a recurrent neural network by changing connections in
the network, resulting in memory maintenance through persistent activity. However, in
biological networks, a large diversity of time-dependent intrinsic mechanisms are also
available. As an alternative to persistent neural activity, we find that learning to make use
of internal adapting dynamics better matches both the observed neural activity and

previous publication at: https://figshare.com/collections/Experience_shapes_activity_dynamics_and_stimulus_coding_of_VIP_inhibitory_cells/4858779/1.

**Funding:** The study received funding from the Allen Institute. The funders had no role in study design, data collection and analysis, decision to publish, or preparation of the manuscript.

**Competing interests:** The authors have declared that no competing interests exist.

behavior of animals in this simple task. The presence of a large diversity of temporal traces could be one of the reasons for the diversity of cells observed. We believe that both learning to keep representations of relevant stimuli in persistent activity and learning to make use of intrinsic time-dependent mechanisms exist, and their relative use will be dependent on the exact task.

## Introduction

Short-term memory, or the ability to temporarily store and maintain task-relevant information, is a fundamental cognitive function. A large body of experimental and computational work suggests that this information can be maintained in persistent neural activity arising from local recurrent connections [1] or even cortical-subcortical loops [2] (for a recent review, we refer the reader to [3]). Both sustained and sequential forms of persistent activity have been observed across tasks and brain regions [4], including the prefrontal cortex (PFC) [5–7].

An alternative mechanism for the maintenance of short-term memories is via short-term synaptic facilitation, using presynaptic residual calcium as a memory buffer [8, 9]. However, recent experimental work suggests that the excitatory synapses in early sensory areas such as mouse primary visual cortex are predominantly depressing [10–13]. Synaptic depression can shape visual processing by providing temporal context as to whether a stimulus was recently experienced (resulting in lower postsynaptic activity) or is novel (resulting in higher postsynaptic activity). Adaptation at the synaptic level or intrinsic firing rate adaptation [14, 15] may then also play an important functional role in the brain.

Here, we study short-term memory using a visual change detection task involving a memory delay [16]. The task is to report whether a stimulus presented before the delay period is the same as the stimulus presented after the delay period. Task difficulty can be modulated by changing the duration of the delay period or the identity of the compared stimuli. In humans, the ability to detect changes in static, flashed scenes is surprisingly poor, giving rise to the phenomenon of "change blindness" [17]. In mice, previous studies using the change detection paradigm mainly presented parametric stimuli such as gratings [18]. In our study, the stimuli consist of flashed natural images (Fig 1). A hallmark of the recorded neural responses in these studies is suppression to repeated stimuli and enhancement to novel stimuli, which is consistent with findings related to mismatch negativity [19] and stimulus-specific adaptation [14]. In the auditory cortex, a recurrent network model with synaptic depression has been shown to explain many of the effects of stimulus-specific adaptation [20].

We use computational models to study the form of short-term memory used in the task (Fig 2). Specifically, we compare the output of a recurrent neural network in which the memory trace is stored in persistent neural activity, with a feedforward neural network with depressing synapses in which the memory trace is stored in short-term synaptic efficacies. We find that the purely feedforward model with depressing synapses can better capture the observed adaptation in neural responses across repeated stimuli, as well as the asymmetric pattern of behavioral responses to different image changes in mice. This adaptation model also makes specific predictions for neural and behavioral responses when stimuli are omitted from a repeated image sequence, which is consistent with the observed data. While there is undeniable evidence that in other tasks short-term memory can be stored in persistent neural activity, our study suggests that short-term synaptic depression can also provide neural circuits with a useful form of short-term memory on the order of seconds. Such a flexibility can be beneficial in term of speed of learning (S7 Fig).

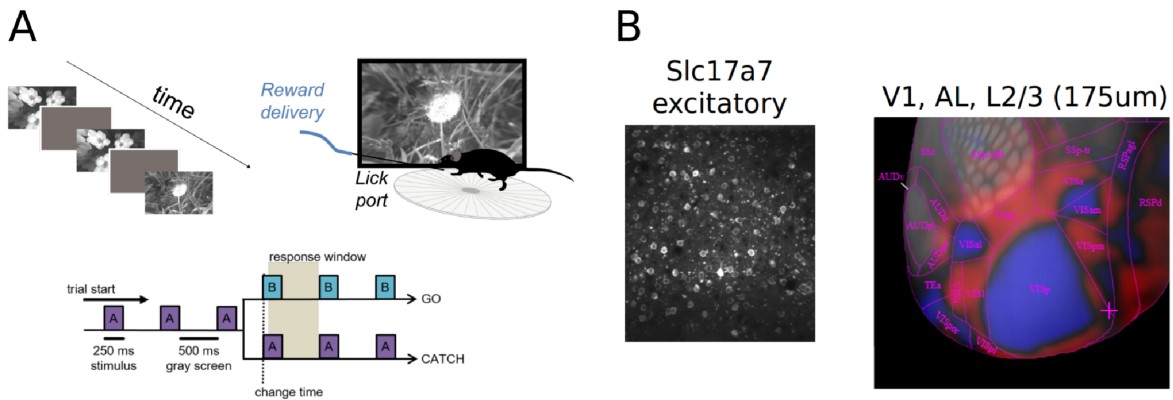

**Fig 1. Overview of the visual change detection task and experiments [16].** (A) Schematic of the change detection task. Mice are trained to detect changes in a sequence of flashed natural images, giving rise to a task structure involving go and catch trials. When the image identity changes (A→B), mice must lick within the response window in order to receive a water reward. When the image identity remains the same (A→A), mice must withhold licks. Images were presented for 250 ms followed by a 500 ms gray screen. (B) During the task, two-photon calcium imaging was used to simultaneously record neural activity from excitatory cells using the pan-excitatory Cre line Slc17a7. An example field of view with labeled cells is shown on the left. All analyses of the experimental data were restricted to mouse visual areas VISp and VISal (layer 2/3). An intrinsic imaging map of mouse visual areas is shown to the right.

## Methods

### Ethics statement

All experiments and procedures were performed in accordance with protocols approved by the Allen Institute Animal Care and Use Committee.

### Visual change detection task

A cohort of mice were trained to perform a visual change detection task involving natural images. Training was performed in a standardized manner, using an automated protocol in which mice progressed through a series of training stages involving static gratings, then flashed gratings, and finally natural images [16]. Stimuli were spherically warped to account for the variable distance from the eye to the periphery of the display monitor (all figures show the unwarped stimuli for ease of visualization). A set of eight natural images ("image set A") was presented, with all images being drawn from a larger set of images previously used as part of the Allen Brain Observatory [21]. Natural images were presented in grayscale, contrast normalized, and matched to have equal mean luminance. The mice were trained on image set A, and had to respond by licking whenever the image within a repeated sequence of images changed (Fig 1). Each image was presented for 250 ms, followed by a blank inter-stimulus interval of mean luminance gray for 500 ms. A small percentage (5%) of image presentations were randomly chosen to be omitted, with omissions never preceding an image change (go trials) or sham image change (catch trials). Importantly, these omissions were only shown during imaging sessions, but never during training. The number of repeat presentations of an image was drawn from a truncated exponential distribution, set to a minimum of 4 repeats and a maximum of 11 repeats. Correct responses were rewarded with water, while premature licks were penalized with a "time-out" period where the trial logic timer was reset. During the task, neural activity was recorded from transgenic mice expressing GCaMP6f in excitatory neurons (Slc17a7-IRES2-Cre; CamK2-tTA; Ai93) in layer 2/3 of primary visual cortex (VISp) and one higher visual area (VISal) using two-photon calcium imaging [16]. All analyses are performed on imaging sessions with simultaneously collected behavioral and physiological data. We

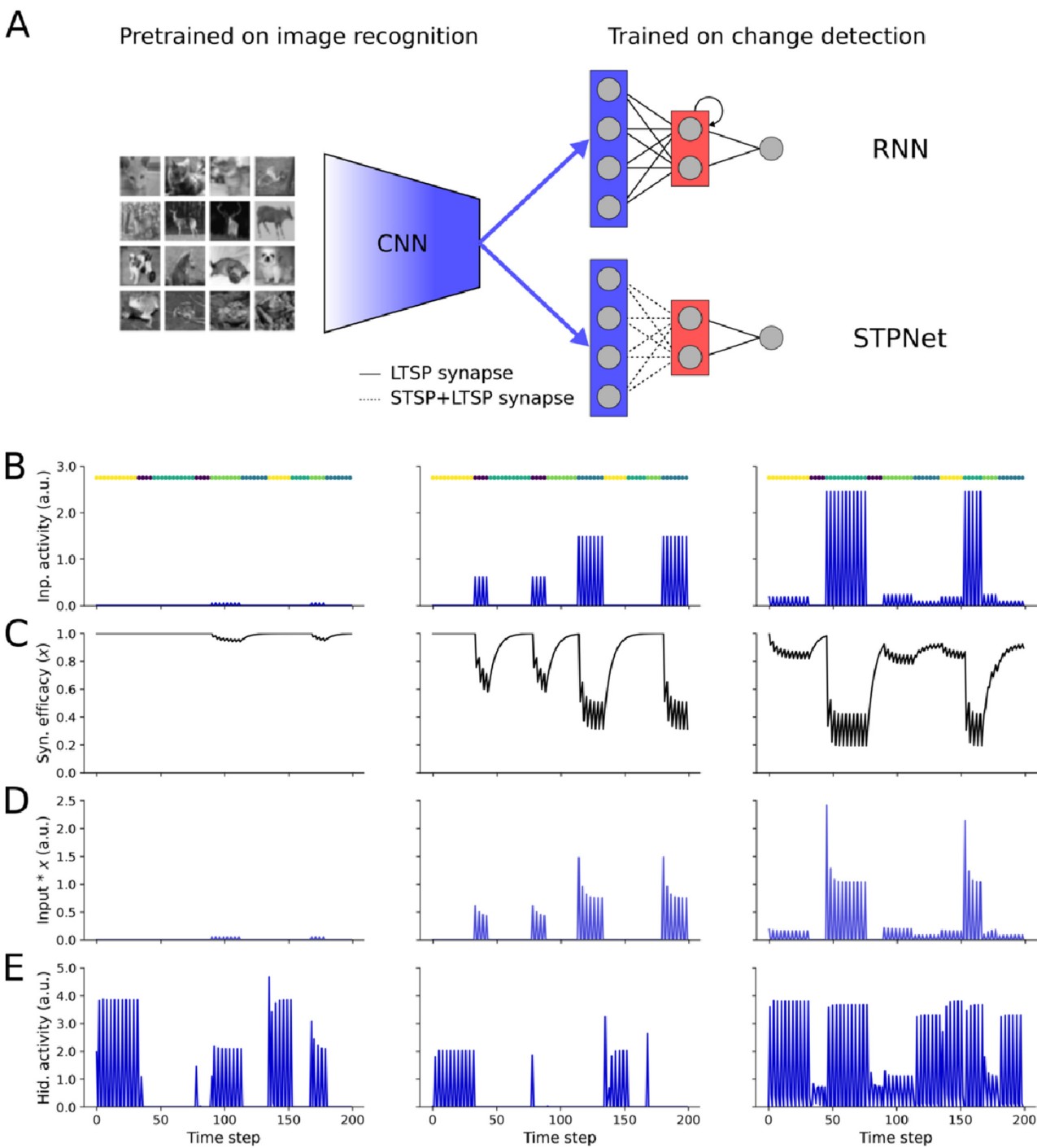

**Fig 2. Overview of the models tested.** (A) Image features are derived from the last fully-connected layer of a convolutional neural network (CNN) pre-trained on a grayscale version of the CIFAR-10 image recognition task. This encoder network maps an input image to a lower-dimensional feature space, which serves as input to the models to the right. Two models of short-term memory were tested based on persistent neural activity (RNN, top) or short-term synaptic plasticity (STPNet, bottom). Models consisted of three layers, roughly corresponding to sensory, association, and motor areas. Model weights were trained with backpropagation (LTSP synapses, solid lines), with the input synapses in STPNet also being modulated by short-term synaptic depression (STSP synapses, dotted lines). (B) Input activity of three example units during the change detection task. Images were presented for 250 ms (one time step) followed by a 500 ms gray screen (two time steps). The left unit ends up responding to only one image and weakly. The center unit responds to two images, one stronger than the other. The right unit responds to four images, in a graded fashion. Image presentation times are color-coded and shown above each plot. (C) Input-dependent changes in synaptic efficacy (depression) for the units from STPNet shown in panel C. (D) Input activity modulated by short-term depression for the units from STPNet shown in panel C. (E) Activity of three example hidden units from a recurrent neural network model, which show more persistent activity. The original images in panel A are reproduced from the CIFAR-10 dataset [22].

combined the data from VISp and VISal for all analyses, as earlier work did not observe major differences between these two areas.

## Neural network models

We compared two models of short-term memory– one based on persistent neural activity (RNN), and another based on short-term synaptic plasticity, specifically synaptic depression (STPNet). For the RNN model, the network was a recurrent neural network with a hidden layer of sixteen neurons connected to a single output neuron representing the decision variable. As the RNN model was able to solve the task, we did not pursue more complex architectures such as LSTMs. For the STPNet model, the network was a two-layer neural network with a single hidden layer of sixteen neurons connected to a single output neuron. In the supplement, we also introduce the STPRNN model, which combines the STPNet and RNN models with one minor modification (S7 Fig). In addition to the set of adapting input synapses, the model has an additional set of non-adapting input synapses, replicating the information that is present in the RNN model. When the weights of the adapting input synapses are set to zero, the model is equivalent to the RNN model, and when the weights of the non-adapting input synapses and recurrent connections are set to zero, the model is equivalent to the STPNet model. For both models, we used the same feedforward neural network to embed the set of natural images into a lower-dimensional feature space (Fig 2). To this end, we first trained a convolutional neural network with two convolutional layers and two fully-connected layers on the CIFAR-10 dataset [22]. Model architectures and number of parameters are shown in Table 1. Images were first converted to grayscale and normalized to the range [0, 1], followed by mean subtraction and division by the standard deviation of the dataset (mean: 0.479, std: 0.239). The model achieved an average accuracy of 62.98 ± 0.78% on the test set. As we were not concerned with state-of-the-art performance on the image classification task, we did not perform an extensive hyperparameter search and only trained this encoder network for ten epochs using stochastic gradient descent with a learning rate of 0.01 and a momentum value of 0.9. After training, we froze the weights of this network, and used the output of the last fully-connected layer before the classifier as input to both the RNN and STPNet models. Although the learned weights in our models can be both positive and negative, our models are meant to only capture the responses of excitatory neurons in the experiments, and here we do not explicitly consider the role of inhibitory neurons. When applied to the change detection task, input images were first resized to 32x32 pixels followed by the same pre-processing as above. Importantly, the input images used were not selected from the CIFAR-10 dataset. To ensure non-negative unit activations within the encoder network, we used the rectified linear nonlinearity at the output of each layer of the network. Furthermore, to make our model more biologically plausible, we also introduced independent multiplicative Gaussian noise (mean = 0, std = 0.5) to the output of the encoder network and to the hidden units to make the responses

**Table 1. Model architectures used for the experiments.** STPNet is the model with short-term synaptic adaptation and RNN is the recurrent neural network. Convolutional layers are denoted as "conv<receptive field size>-<number of channels>". "maxpool" denotes max pooling using a 2x2 window and a stride of 2. "FC" denotes fully connected layers with the given number of units. "RC" denotes recurrent layers with the given number of units. The ReLU activation function is not shown for brevity. The shared pretrained feature extractor layers are shown in italics (63658 total params).

| Model | Network architecture | | | | | | | | | Params |
|---|---|---|---|---|---|---|---|---|---|---|
| STPNet | *conv5–8* | *maxpool* | *conv5–16* | *maxpool* | *FC-128* | *FC-64* | FC-16 | FC-1 | sigmoid | 64715 |
| RNN | *conv5–8* | *maxpool* | *conv5–16* | *maxpool* | *FC-128* | *FC-64* | RC-16 | FC-1 | sigmoid | 64971 |
| STPRNN | *conv5–8* | *maxpool* | *conv5–16* | *maxpool* | *FC-128* | *FC-64* | FC/RC-16 | FC-1 | sigmoid | 65995 |

non-deterministic. All models were trained and simulated using the Pytorch neural network framework (v. 1.1.0) [23].

## Short-term synaptic plasticity

For the model with short-term synaptic plasticity (STPNet), we included dynamics of short-term depression at the synapses from the output of the encoder network (Fig 2). We emphasize that this plasticity is presynaptic and non-Hebbian, but can interact with the weight updates from backpropagation during neural network training, similar to the idea of "fast weights" from machine learning [24]. We modeled short-term depression based on the model proposed by Tsodyks and Markram [25], but instead of using spikes, we used a firing rate approximation in terms of the activations of units within the network:

$$\frac{dx(t)}{dt} = \frac{1 - x(t)}{\tau_x} - Ux(t)r(t) \tag{1}$$

where $x(t)$ represents the fraction of neurotransmitter available, $U$ represents the calcium concentration in the presynaptic terminal, $r(t)$ is the presynaptic activity at time $t$ and $\tau_x$ is the recovery time constant of the synapse. The total amount of input at the postsynaptic unit at time $t$ becomes:

$$I(t) = Wx(t)r(t) \tag{2}$$

where $W$ is the synaptic efficacy of the synapse without short-term synaptic plasticity. For simplicity, all synapses from the same presynaptic unit were assumed to be modulated similarly. Unless otherwise noted, we fix $U$ to be 0.5 with a $\tau_x$ of 1.5 seconds, corresponding to a regime where short-term depression dominates [13]. The exact choice of $U$ and $\tau_x$ did not qualitatively change our results. We show example training trajectories for STPNet models with different $\tau_x$ values and delay periods in S8 Fig. We discretized time with a time step $\Delta t$ of 250 ms, corresponding to the duration of each stimulus presentation during the experiments. We solved for Eq 1 using the implicit backward Euler method.

## Neural network training

To model the mice's behavioral responses, we treat the change detection task as a binary classification problem where the network has to determine whether the presented image is a change image (represented by the label 1) or a repeat image (represented by the label 0). To prevent the model from being over-confident in its predictions and to capture the inherent behavioral variability in mice, we sampled the model's responses from a Bernoulli distribution based on the sigmoid of its output, which was bounded between 0 and 1. The models were trained using sequences of images drawn from the same set of images and distribution of change times as those used in the experiments. The models were trained using stochastic gradient descent and the ADAM optimizer [26] with the default settings (learning rate of 0.001 and beta values of 0.9, 0.999), using a binary cross entropy loss with a positive weight of 5. We trained a total of 10 networks for each type of model with different random seeds as model initializations. All layers were initialized using He uniform initialization [27]. All reported results, unless otherwise noted, are averages over the trained networks. We masked the loss during inter-stimulus intervals (so that only model responses on image presentations contributed to the loss), and we also used an additional L2 penalty with a weight of 0.001 on the activations of hidden units to encourage low levels of activity. This is commonly done when training recurrent neural networks [28, 29], as cortical firing rates are generally low. This L2 penalty also causes the recurrent weights in the RNN model to be relatively low in magnitude (S9 Fig). The models were

trained for a maximum of 5000 epochs, with a d-prime early stop criterion of 1.5 (based on the hit and false alarm rates observed in mice). We used a patience of 5, meaning the d-prime value had to be above the d-prime early stop criterion for at least 5 epochs before training stopped. The exact choice of model hyperparameters did not qualitatively change our results.

## Analyses of experiment and model data

**Quantifying response asymmetry.** We quantified behavioral performance using the response probability matrix, which captures the pattern of responses to each pair of image transitions (Fig 3C). In this matrix, catch trials (trials where no image change occurred) are along the diagonal and go trials (trials where an image change occurred) are off the diagonal. For an image set of eight natural images, there are a total of 64 possible image transitions. The response probability matrix was computed for both the experiments and the models. We quantified similarity between the model and mice response probability matrices using the Pearson correlation coefficient. In addition, we quantified the degree of symmetry in the response probability matrices using the following metric $Q$:

$$Q = \frac{\|M_{sym}\| - \|M_{anti-sym}\|}{\|M_{sym}\| + \|M_{anti-sym}\|} \tag{3}$$

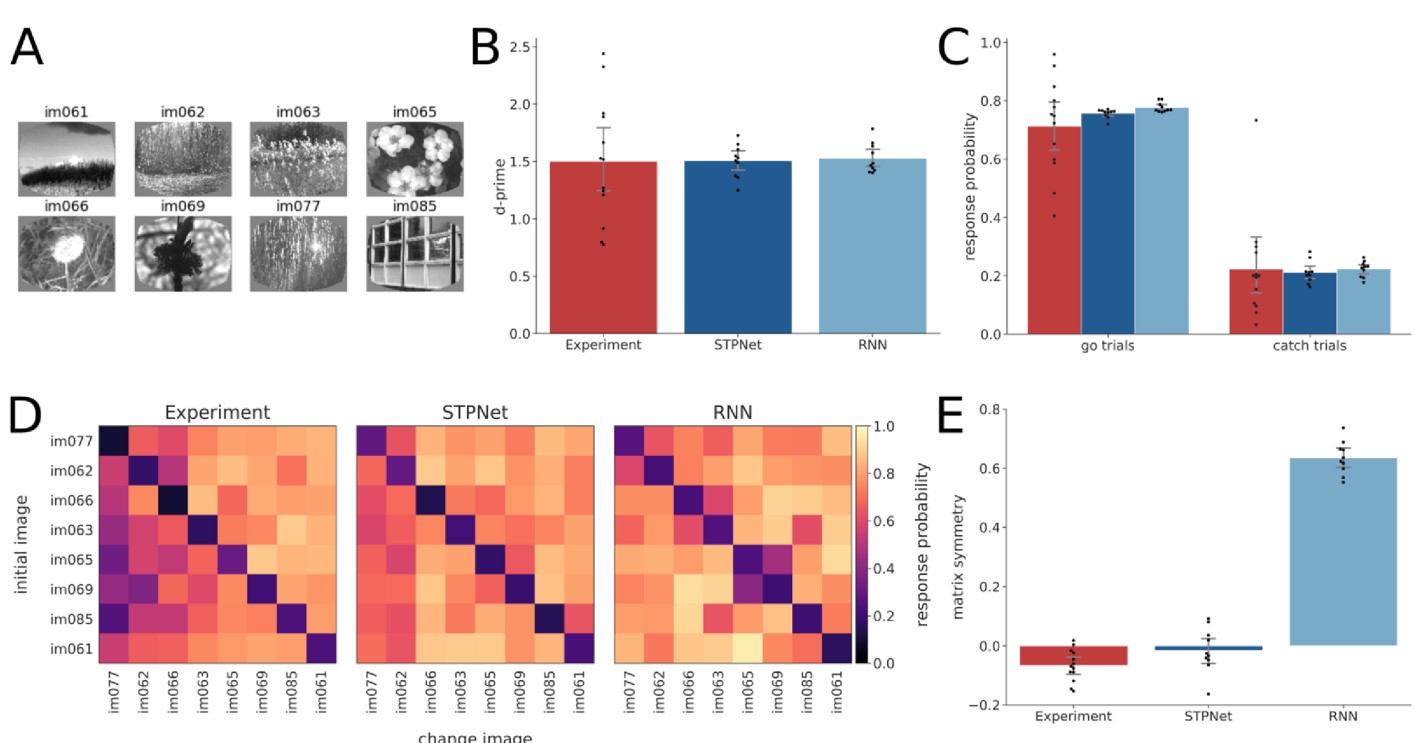

**Fig 3. Quantification of behavioral responses in the experiments and the models.** (A) Set of eight images that were used during the task. (B) D-prime values computed for the experiments (red) and the two models (blue). Error bars show 95% confidence intervals on the mean. The same color convention is used throughout all figures. (C) Hit rate (left) and false alarm rate (right) for the experiments and the models. Error bars show 95% confidence intervals on the mean. (D) Average response probability matrices which show the probability of responding to each of the possible 64 image transitions during the task (go trials off the diagonal, catch trials along the diagonal). The experimental response probability matrix shows an asymmetry in the detectability of images, which is partly captured by the STPNet model, but not the RNN model. (E) Quantification of response probability matrix asymmetry. Matrix symmetry is bounded between −1 and 1, with negative values indicating asymmetry and positive values indicating symmetry. The experiments and the STPNet model show a small degree of matrix asymmetry, while the RNN model shows high matrix symmetry. Error bars indicate 95% confidence intervals on the mean.

$$M_{sym} = 1/2 \; (M + M^T) \tag{4}$$

$$M_{anti-sym} = 1/2 \; (M - M^T) \tag{5}$$

where $M$ is the mean-subtracted response probability matrix (ignoring the elements along the diagonal), $M^T$ is the transpose of this matrix, $M_{sym}$ and $M_{anti-sym}$ are the symmetric and anti-symmetric components of the matrix $M$, respectively, and $\|.\|$ represents the Frobenius norm. This metric is bounded between -1 and 1, with more symmetric matrices having values closer to 1 and more anti-symmetric matrices having values closer to -1.

**Change modulation index.**   One possible signature of neural adaptation is stronger responses for change images compared to repeat images. To test this hypothesis, we computed the change modulation index for each neuron by comparing mean responses to change and pre-change images (i.e. the images presented on go trials and the images presented just before a change image, respectively). The change modulation index (CMI) is defined as:

$$CMI = \frac{r_{change} - r_{pre-change}}{r_{change} + r_{pre-change}} \tag{6}$$

where $r_{change}$ is the mean response to change images and $r_{pre-change}$ is the mean response to pre-change images. The change modulation index is bounded between -1 and 1, with neurons that respond more strongly to change images having values closer to 1 (suggesting adaptation) and neurons that respond more strongly to pre-change images having values close to -1 (suggesting facilitation). We show a schematic of how the change modulation index is computed in S1 Fig. In our analyses, we excluded neurons that had a negative mean dF/F response to either the change or pre-change images, which would result in change modulation indices outside the range [-1, 1]. Alternate modulation indices based on either comparing the mean response to go versus catch trials, or the mean response to change images versus repeat images a fixed number of repeats later, gave qualitatively similar results.

**Decoding analyses.**   We also performed decoding analyses to determine what task-relevant information could be reliably decoded from the population of recorded neurons. Specifically, we trained linear support vector machines (SVMs) to decode either image identity on each image presentation (multi-class classification) or whether a given image was a change or pre-change image (binary classification). We used change and pre-change images to avoid potential problems with imbalanced data when using go and catch trials. We performed three-fold cross-validation, and report mean decoding accuracy averaged across the three folds. For each experimental session, we used all recorded neurons in our decoding analyses, only excluding neurons which had a negative mean dF/F response to either the change or pre-change images. We also tested the effect of number of neurons on mean decoding accuracy by randomly sampling a fixed number of neurons from each session ($N = 10$ cross-validated samples) and training the decoders using this smaller subset of neurons (S2 Fig). To understand whether neurons which showed greater change modulation were more important for change decoding, we computed the Pearson correlation coefficient between the weights from the change decoder and the corresponding change modulation indices for each neuron. We also performed a similar analysis on the hidden units from the two models, using the hidden unit to output unit weights as a measure of change decoding (S3 Fig). To assess whether the trained decoders were predictive of mice's behavioral responses, we used the Jaccard score to quantify the similarity between the decoder's predictions and the mice's actual choices. Importantly, while the decoders were trained on change and pre-change images, the decoders were tested

on go (equivalent to change images) and catch trials. As a control, we shuffled the decoder's predictions ($N$ = 1000 times) and computed the Jaccard score between the shuffled predictions and the mice's choices, which gives a measure of similarity based on random chance alone.

**Dimensionality reduction.** To visualize the structure of population activity during the change detection task, we also performed PCA dimensionality reduction on the activity of model units and neurons recorded in the experiments (Fig 5). We retained those principal components (PCs) which captured 95% of the variance of the neural activity, the number of which could be different for each experiment. We then computed a Euclidean distance metric using either the neural activities themselves (which is equivalent to using all PCs) or using a subset of the PCs approximating a low-dimension projection of the data (typically just the first PC). We computed this distance from the origin as a function of the number of image repeats. Similar results were obtained using linear discriminant analysis (LDA) or isomap as dimensionality reduction techniques (S5 and S6 Figs).

**Omitted image presentations.** A small percentage (5%) of image presentations were randomly omitted during the experiments. In addition to the response probability on go and catch trials, we quantified the response probability to these omitted images, as well as to the post-omitted images (defined as image presentations that came after the omission). For the two models, we conditioned the responses to post-omitted images on the omitted image response, i.e. if the model responded to the omitted image, its response probability on the post-omitted image presentation was always defined to be zero. This was not required for the experiments due to existing trial logic which aborted trials on the omitted image if the mouse responded prematurely.

## Results

### Visual change detection task

Mice were trained on a go/no-go visual change detection task with natural images [16]. In this task, images are presented repeatedly in a periodic manner (250 ms stimulus presentation followed by 500 ms gray screen). On 'go' trials, the image identity changes and mice have to report the change by licking within 750 ms in order to receive a water reward (Fig 1A). 'Catch' trials when the image identity does not change are used to quantify false alarms. After training, two-photon imaging was performed in primary visual cortex (VISp) and one higher visual area (VISal) using transgenic mice expressing the calcium indicator GCaMP6f in excitatory pyramidal cells (Slc17a7-IRES2-Cre; CaMKII-tTA; Ai93-GCaMP6f, Fig 1B). As a previous study did not observe major differences between these two visual areas [16], we also combined these areas for the analyses reported here (see Methods for details). This resulted in a total of 11 mice and 13 imaging sessions being included in our analyses. For a more detailed description of the task and the corresponding dataset, we refer the reader to [16].

### Asymmetry in the detectability of natural images

We tested two types of models in the context of the change detection task, one based on persistent neural activity and one based on short-term synaptic depression (Fig 2). We first used a convolutional neural network pre-trained on an image classification task to generate low-dimensional image features. The resulting input activity to the models was sparse, matching the observed responses in a large-scale survey of mouse visual cortex [21]. We then trained two types of models: a recurrent neural network (RNN) and a feedforward neural network with short-term synaptic plasticity (STPNet). The input layer of the models consisted of 64 units, followed by a hidden layer of 16 units, and a single output unit representing the decision variable. For the RNN model, units in the hidden layer were recurrently connected to each

other. For the STPNet model, we only considered depressing synapses from the input layer to the hidden layer (Fig 2A; see Methods). All units were linearly rectified to ensure non-negative activations. Example responses of model units are shown in Fig 2B–2E.

We trained 10 networks to solve the visual change detection task, in which the networks had to indicate whether sequentially presented stimuli were the same or different (a binary classification task). All model weights were updated using the backpropagation algorithm. To prevent the models from overfitting, we used early stopping based on the average d-prime criterion of the mice in the study. Both models reached similar performance as the mice in the experiments (Fig 3A–3C). There were no statistically significant differences between model and experiment d-prime values as determined by one-way ANOVA ($F_{(2, 30)} = 0.018$, $p = 0.98$). Similarly, we did not observe statistically significant differences between model and experiment hit rates ($F_{(2, 30)} = 1.198$, $p = 0.32$) or false alarm rates ($F_{(2, 30)} = 0.037$, $p = 0.96$). We performed a more fine-grained quantification of behavior by computing response probability matrices, which measure the probability of responding to each image transition (Fig 3D). Off-diagonal elements of this matrix represent go trials, while diagonal elements represent catch trials. As expected, both the mice and the models respond with higher probability on go trials versus catch trials. We also observed that on average mice show a striking behavioral asymmetry in their responses, with certain image transitions being either easier or harder to detect. Here, the asymmetry is with respect to the diagonal of the response probability matrix, where the response probability for the transition from image $i$ to image $j$ is different than image $j$ to image $i$ (for example, compare the difference in response probabilities for the image transition at the top-right and bottom-left of the matrix).

We quantified this asymmetry by computing a matrix symmetry value $Q$ for each response probability matrix (see Methods for more details). To facilitate comparison between the experiments and the models, the response probability matrices are always ordered by the mean detectability of images across mice in the experiments. We found a statistically significant difference between model and experiment matrix symmetry values as determined by one-way ANOVA ($F_{(2, 30)} = 427.08$, $p < 0.001$). A Tukey post hoc test revealed that the matrix symmetry values were statistically significantly higher for the RNN model ($0.63 \pm 0.05$) compared to the experiments ($-0.07 \pm 0.05$, $p = 0.001$) and STPNet model ($-0.02 \pm 0.07$, $p = 0.001$). There was no statistically significant difference in the matrix symmetry values between the experiments and STPNet model ($p = 0.13$). These results indicate that the RNN model showed a more symmetric response probability matrix. We find that the STPNet model also shows a higher Pearson correlation of its response probability matrix with the experimental response probability matrix ($R = 0.79$, $p < 0.001$). On the other hand, the RNN model showed a weaker correlation with the experimental response probability matrix ($R = 0.65$, $p < 0.001$). Although both models are able to perform the the change detection task, we find that the STPNet model better matches the observed behavioral data (Fig 3E).

## Adaptation as a hallmark of neural and model responses

One hallmark of the recorded neural responses is strong adaptation, with many neurons showing reduced responses to repeated image presentations (Fig 4A). We quantified this adaptation by computing a change modulation index for each neuron (see S1 Fig and Methods for details). The change modulation index compares the mean response of the neuron on the first presentation of a change image (i.e. a go trial) with the mean response of the neuron on the corresponding pre-change image (Fig 4B and 4C). The change modulation index is bounded between -1 and 1, with positive values indicating higher responses to change images versus repeat images (adaptation) and negative values indicating higher responses to repeat images

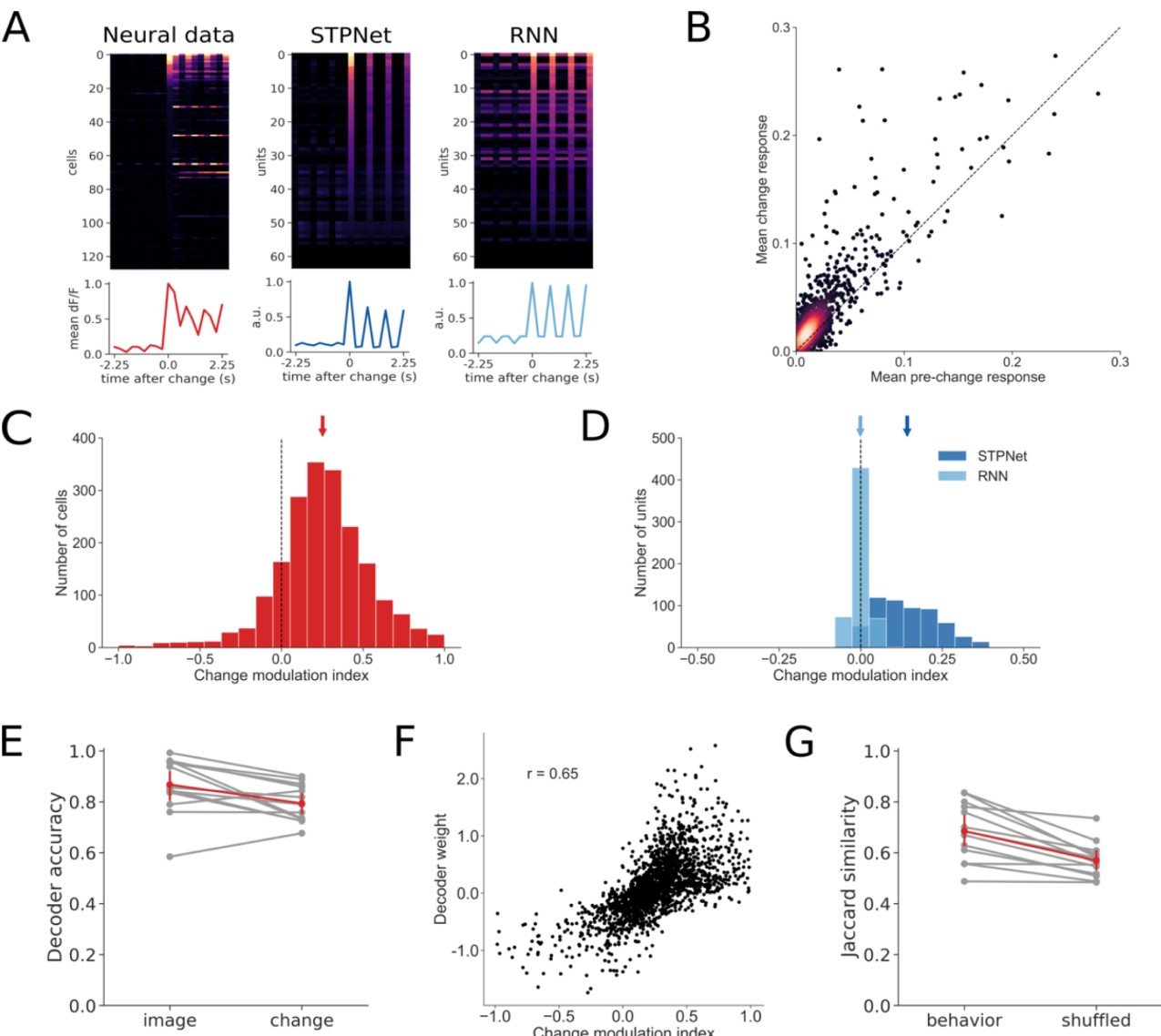

**Fig 4. Quantification of change modulation in the experiments and the models.** (A) Example responses centered around the change image time, for the experiments and the two models. The change image was chosen to be the preferred image for either the cell or the model unit under study. The line plots below show the population average time course. (B) A scatter plot of the mean dF/F response on change images versus the mean dF/F response on pre-change images for all neurons recorded in the experiments. Points above the unity line indicate larger mean response on change images versus pre-change images. (C) Quantification of the change modulation index (defined in Methods) in the experiment for the cells shown in panel (B). The mean change modulation index for all cells is positive (mean = 0.25, sem = 0.01) and significantly different than zero ($p < 0.001$, Wilcoxon signed-rank test). (D) Quantification of the change modulation index for units in the first layer of the two models. The STPNet model shows a positive mean change modulation index (mean = 0.14, sem = 0.004), while the RNN model shows a change modulation index close to zero (mean = 0.0, sem = 0.001). The change modulation index is significantly different than zero for the STPNet model ($p < 0.001$), but not the RNN model ($p = 0.26$). (E) Mean three-fold cross-validated decoding accuracy for image identity and image change across sessions ($N = 13$). Both image identity and image change were decodable above chance from population activity. Chance accuracy is 12.5% for image identity and 50% for image change. (F) Scatter plot of average decoder weights and change modulation indices for the neurons recorded in the experiments. A strong positive correlation ($r = 0.65$) is evident, suggesting that neurons which adapt are informative for change detection. (G) Jaccard similarity between decoder predictions and actual mouse behavior. The decoder predicts mouse behavior better than chance level, as determined by a shuffle control.

versus change images (facilitation). Neurons showed positive change modulation indices (mean = 0.25, sem = 0.01) that were significantly different than zero ($p < 0.001$, Wilcoxon signed-rank test), indicating larger responses to change images versus repeat images. We also computed change modulation indices for the input units of the two models, and found that only the STPNet model showed significant positive change modulation indices (mean = 0.14, sem = 0.004, $p < 0.001$, Wilcoxon signed-rank test). In contrast, the RNN model had change modulation indices that were not significantly different than zero (mean = 0.0, sem = 0.001, $p = 0.26$, Wilcoxon signed-rank test). The change modulation indices are slightly higher for the experiments than for the models (Fig 4C and 4D, note the difference in scales along the x-axis). There was a statistically significant difference in the distribution of experimental change modulation indices compared to those of the STPNet ($p < 0.001$) and RNN ($p < 0.001$) models by a 2-sample Kolmogorov-Smirnov test. This suggests that the models are not able to fully capture the neural mechanisms of change modulation due to the complexity of the biology. There was also a statistically significant difference in the distribution of change modulation indices between the STPNet and RNN models ($p < 0.001$). Given that both models are able to solve the change detection task, these differences in the change modulation indices suggest different representations of short-term memory being used by the two models.

To further understand what task-relevant information is encoded during the change detection task, we performed a set of decoding analyses using the recorded populations of neurons from primary visual cortex (VISp) or one higher visual area (VISal). Both image identity and image change were decodable from population activity above chance in the experiments (Fig 4E). Mean decoding accuracy was 86.7% and 79.4% for image identity and image change, respectively; chance level was 12.5% for decoding image identity and 50% for decoding image change. The number of recorded neurons per experiment ranged from 57 to 310 (mean = 157, std = 77). Mean decoding accuracy for image identity and image change both saturated after including approximately 100 neurons (S2 Fig). Interestingly, we observed a strong positive correlation ($R = 0.65$, $p < 0.001$) between the change decoder weight and the change modulation index for each neuron (Fig 4F). These results suggest that neurons which showed greater adaptation were more informative of image change, and their responses to the change image could be used to decode whether the presented image was a change image or a repeat image. We performed a similar analysis using model hidden units (which are directly connected to the output neuron), and found that both models also showed a strong positive correlation between the output weights and the change modulation indices (S3 Fig). Finally, we wanted to understand if the trained change decoders were predictive of the mice's actual behavior on a trial-to-trial basis. We quantified the agreement between each change decoder's predictions and the mouse's actual responses by computing a Jaccard score, which measures the degree of similarity between two binary vectors. Using a shuffle control, we found that the change decoder showed greater agreement with the mice's behavior than that simply expected by chance (69.1% true versus 57% shuffled, Fig 4G).

## Low-dimensional analyses of neural and model activity

To better understand the structure and dynamics of population activity during the change detection task, we performed PCA dimensionality reduction on the population of recorded neurons from VISp or VISal and studied how this representation changed over the course of a trial. We found that the first two principal components captured on average 50.4% of the variance in VISp and 44.3% of the variance in VISal. We found that the representations of images cluster to different regions of the low-dimensional space (Fig 5A). We also find that repeated presentations of the same image pulls the representations closer to the origin of the PCA space

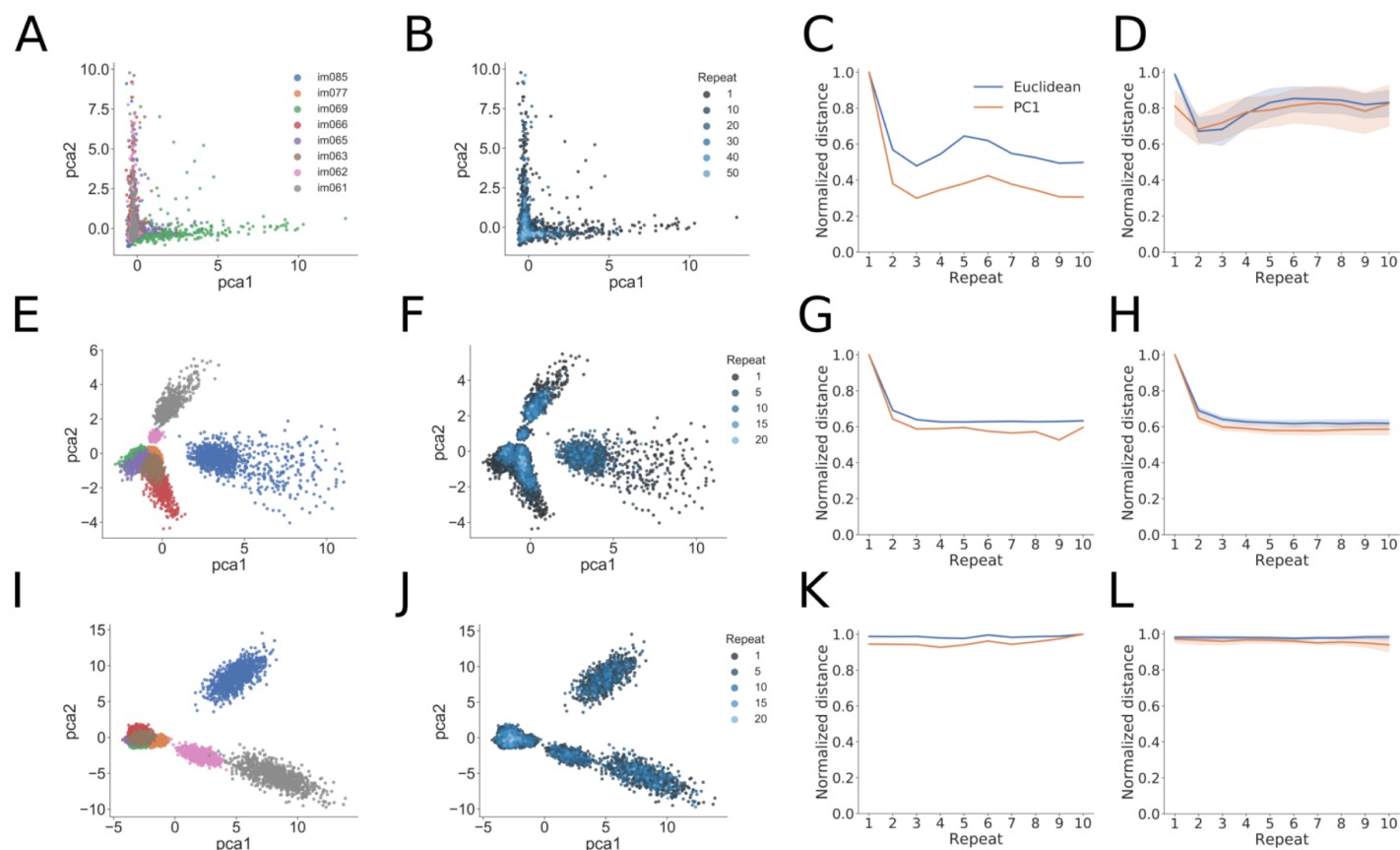

**Fig 5. Low-dimensional analyses of population activity for an example experiment and trained examples of the two computational models.** (A) PCA analysis showing projection of population activity onto the first two PCs. Colors label the identity of the shown image and are the same for panels E and I below. The different images can be well separated in this low-dimensional space. (B) The same projection as in panel (A), with the color now showing the number of image repeats (more blue hue indicates higher repeats). Note the difference in number of repeats when compared to panels F and J below. The experiments show a "collapse" toward the origin of the low-dimensional space with increasing number of repeats. (C) Quantification of the population-level adaptation effect as a function of number of repeats. The distance is plotted using either using the Euclidean distance of the full population activity (blue) or the absolute value of the first PC in the low-dimensional space (orange). The experiment shows decreasing distances with repeats, and the fact that the first PC tracks the Euclidean distance indicates a low-dimensional effect. (D) Same convention as in panel (C), now showing the measured distances over all experimental sessions ($N = 13$). The mean and 95% confidence intervals are shown for each distance plot. (E-H). Same as (A-D), but for the STPNet model, which shows similar trends as for the experiments. (I-L) Same as (A-D), but for the RNN model. The RNN model shows differences with the experiments and STPNet model, mainly a constant distance as a function of image repeats.

(Fig 5B). We quantified this effect by calculating a distance from each image representation to the center of the PCA space (see Methods for details). We did this using either the Euclidean distance of the full population activity, or a low-dimensional view of this activity by using only a subset of the principal components. For the experiments, we observed a decrease in distance from the origin with increasing number of image repeats, which is consistent with the adaptation effect observed at the single-neuron level (Fig 5C). In many cases, the first principal component was sufficient to capture this effect, suggesting a low-dimensional correlate of image repetition. This effect was reproducible across multiple mice and visual areas (Fig 5D).

For the STPNet model, we also find that the different images map to different regions of the PCA space, and that this representation moves closer to the origin with increasing image repetitions (Fig 5E and 5F). In contrast, for the RNN model, we found that the representations largely cluster together based on image identity, and do not move significantly with increasing image repetitions (Fig 5J and 5K). The distance from the origin decreases rapidly for the STPNet model, which is similar as in the experiments, but remains relatively constant for the

RNN model (compare Fig 5G and 5K). These findings held over the ensemble of models that were tested (Fig 5H and 5L). As the origin of PCA space represents the mean across all image presentations, this suggests that the representations of images becomes closer to the gray inter-stimulus interval with increasing number of stimulus repetitions. We hypothesize that the "pulling" of neural activities towards the center of the PCA space may be a useful mechanism to form a clear decision boundary which can be used to determine whether an image is novel or repeated. We show the fraction of variance explained for the experiments and models as a function of the number of PCA components used in S4 Fig. We note that similar results were obtained using other dimensionality reduction techniques (S5 and S6 Figs).

## Models make different predictions on image omissions

During the experiment, a small percentage (5%) of image presentations were randomly omitted, effectively increasing the inter-stimulus duration between image presentations (Fig 6A). In rare cases, two consecutive stimuli could be omitted, further extending the inter-stimulus duration, but we did not explicitly analyze this condition due to the low number of available trials. The presence of stimulus omissions allows us to test the two different mechanisms of short-term memory, which make different predictions under this condition. Models based on persistent neural activity should show sustained activation during the omissions, which is what we observe in the RNN model. In contrast, both the neural data from the experiments and the STPNet model do not show significant activity during the omitted stimulus presentations (Fig 6B).

We did not observe statistically significant differences in the response probability on go trials ($F_{(2, 30)}$ = 1.198, $p$ = 0.32) or catch trials ($F_{(2, 30)}$ = 0.037, $p$ = 0.96). However, we did observe statistically different response probability values when comparing all ($F_{(2, 30)}$ = 37.01, $p < 0.001$), omitted ($F_{(2, 29)}$ = 149.86, $p < 0.001$), and post-omitted stimulus presentations ($F_{(2, 29)}$ = 61.92, $p < 0.001$). A Tukey post hoc test revealed statistically significant differences in

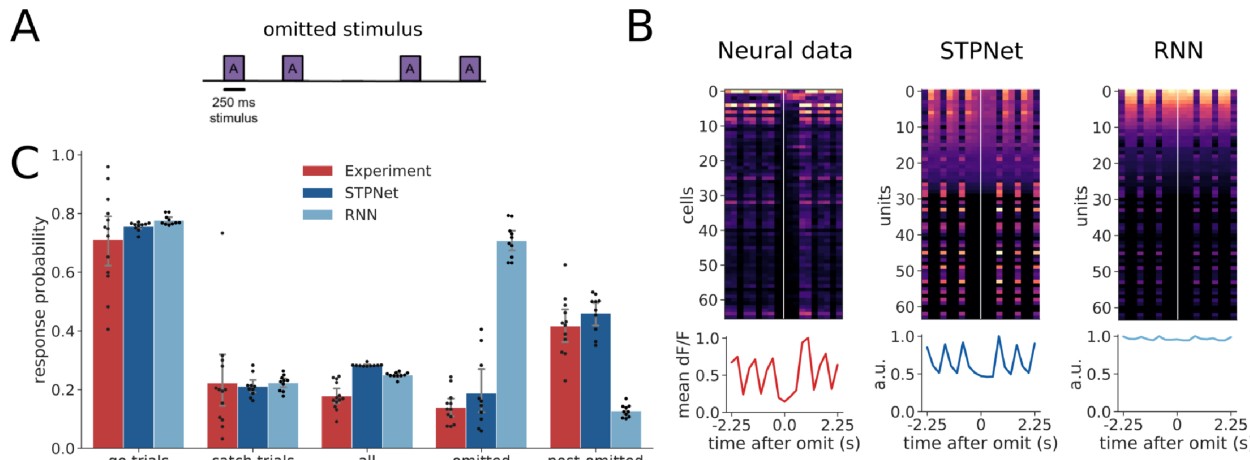

**Fig 6. Quantification of response probabilities in the experiments and models.** (A) Schematic of omitted stimulus presentations during the experiments. A small percentage (5%) of stimulus presentations were omitted during the experiment. (B) Example responses centered around the omitted stimulus presentation time, for the experiments and the two computational models. Cells in the experiments and STPNet did not respond during the omission, while many cells in the RNN model were active during the omission. The line plots below show the population average time course. (C) Average response probabilities for go trials, catch trials, all stimulus presentations, omitted stimulus presentations, and post-omitted stimulus presentations. Response probabilities on go trials, catch trials, and all stimulus presentations are similar across the experiments and models. Both the experiments and STPNet model show an elevated response probability following omission (post-omitted stimulus presentations), while the RNN model shows a high response probability on the omitted stimulus presentations. Error bars indicate 95% confidence intervals.

response probability to all stimulus presentations between the experiments and STPNet ($p = 0.001$), the experiments and RNN ($p = 0.001$), and the STPNet and RNN models ($p = 0.05$). A Tukey post hoc test revealed that the response probability on omitted stimulus presentations was significantly higher for the RNN model ($0.71 \pm 0.06$) compared to the experiments ($0.14 \pm 0.06$, $p = 0.001$) and STPNet model ($0.19 \pm 0.12$, $p = 0.001$). There was no statistically significant difference in omitted response probability between the experiments and STPNet model ($p = 0.37$). Similarly, a Tukey post hoc test revealed that the response probability on post-omitted stimulus presentations was significantly lower for the RNN model ($0.13 \pm 0.02$) compared to the experiments ($0.42 \pm 0.10$, $p = 0.001$) and STPNet model ($0.46 \pm 0.07$, $p = 0.001$). Again, there was no statistically significant difference in post-omitted response probabilities between the experiments and STPNet model ($p = 0.36$). This suggests that the RNN model views the absence of an image as a change in its input, and responds to this change accordingly (note that all models were trained without omissions, and only tested with omissions in line with the experiments). In contrast, both mice and the STPNet model show lower response probabilities for the omitted stimulus presentations, but higher response probabilities following omission. This is consistent with a synaptic depression mechanism in which the increased inter-stimulus interval allows synapses to recover from adaptation, and effectively corresponds to a "decay" in the short-term memory of the previously seen image. The STPNet model makes the prediction then that increased inter-stimulus intervals will result in decreased model performance, specifically in the form of increased false alarm rates.

## Discussion

The current study focuses on analyzing neural and behavioral data from a visual change detection task [16], constructing simplified conceptual models which link neural activity to mouse behavior. These models allowed us to test the predictions of different short-term memory mechanisms. Our results show clear behavioral, neural, and stimulus perturbation evidence that adaptation mechanisms, rather than persistent activity, is used for change detection of natural images. These findings extend previous studies which have shown the importance of adaptation for change detection in the auditory domain [20]. Our study also provides a general road map for relating neural and behavioral data to computational models, whose predictions can then be tested with novel stimuli.

### Understanding the neural mechanisms of short-term memory

Our models suggest that short-term synaptic depression may also serve as a substrate for short-term memory in neural circuits of early sensory cortex, and it is the main memory source in the described change detection task. Multiple lines of evidence support our proposed model with depression (STPNet)– the asymmetric pattern of behavioral responses for mice trained on the change detection task, the adapting profile of neural responses across image repetitions, and the pattern of responses to omitted image presentations. Image repetition causes the input synapses in our model to adapt, reducing information about image identity. Upon presentation of a change image, a new set of input units are activated (release from inhibition). Images with more salient features (e.g. low-frequency, high-contrast edges) will strongly drive the input units in our model, while less salient images will only drive the input units weakly. The level of input unit activity influences the responses of hidden units, which is then used to directly decode image change. As a result, synaptic depression acts as a temporal filter which enables the comparison of repeated and novel stimuli, with stimuli which are different from those encountered in the recent past showing higher activity than for repeated stimuli.

The behavioral asymmetry reflects the fact that different average levels of activations for multiple images (which we refer to as image saliency) causes them to be processed differently. Images with high saliency which change from images with low saliency are easily responded to while images with low saliency changing from images with high saliency are easily missed. This type of behavior is lacking in models which do not have any form of bottom-up attention, such as a CNN followed by an RNN. Models which include adaptation mechanisms can make the strength of the short-term memory be dependent on the saliency of the stimulus to be memorized and thus reproduce the memory asymmetry observed in the data.

In contrast, while a model which stores short-term memory in persistent neural activity was also able to solve the task, the representation used by the model was much less consistent with the constraints given by the observed data. Over all the models that we trained and tested, we empirically found that recurrent neural networks tend to show more symmetric responses. Understanding why these networks show symmetric responses is left for future work. Although we cannot rule out the contribution of persistent neural activity in other cortical areas, our results suggest that in early sensory cortex with the change detection task used here, a mechanism as simple as depressing synapses on the neurons representing the sensory input may be sufficient to capture the neural dynamics associated with behavior. Importantly, future work will require the use of causal optogenetic perturbations to test and confirm the results of our experimental and modeling analyses.

## Comparison to other models

A large body of literature suggests that both persistent neural activity and short-term synaptic plasticity can be used in tasks that involve short-term memory. Attractor-based models suggest that memories can be maintained in persistent neural activity across a recurrently-connected population of neurons. More recently, many studies have shown that recurrent neural networks trained in a task-based manner can reproduce experimentally observed neural dynamics [30]. A related line of research shows that sequential neural activity, where neurons within a population are activated at different time points can also be used to store short-term memories [31–33]. The sustained or sequential nature of the persistent neural code used can vary with the task and other factors [29].

Earlier work on spiking neural networks with both recurrent connections and dynamic synapses showed that they can exhibit a form of fading memory which can be used to solve complex tasks such as speech recognition [34]. Related to our study, it has been shown that recurrent neural networks with synapses modulated by short-term synaptic plasticity can use both persistent neural activity and synaptic efficacies to store short-term memories [28]. Furthermore, these authors found that models were more likely to use persistent neural activity in cases where the short-term memory information has to be manipulated rather than simply maintained, e.g. during a delayed match-to-rotated sample task. Our work differs in that we show that synaptic depression at the input synapses alone can be also used as a bottom-up memory signal (see [20] for a similar computational modeling study in auditory cortex). Importantly, we note that the short-term synaptic depression model we used here is presynaptic and non-Hebbian, corresponding to changes which happen at a much faster timescale than typically observed during learning.

In the experiments, neurons showed varying levels of adaptation (some neurons did not adapt) while in the models with persistent activity, the majority of units showed sustained responses. In non-human primates, there is extensive experimental work demonstrating repetition suppression and match enhancement effects during various visual tasks [6, 35, 36]. To explain some of these findings, several models have been proposed, including fatigue-based

models similar to ours that rely on firing rate adaptation or synaptic depression [14]. Interestingly, [37] show that repetition suppression is always accompanied by match enhancement and persistent activity. Our results are largely consistent with this body of literature. We also note that a small fraction of neurons had a negative change modulation index, indicating slight facilitation. By design, the STPNet model is unable to produce negative change modulation indices, while a smaller fraction of units in the RNN produced negative change modulation indices. Interestingly, in the context of predictive coding, a two-channel code has been proposed, with one channel showing transient responses that exhibit a similar adaptation profile in neurons across repeated stimuli, and the other channel showing sustained responses in a smaller subset of the population across all image presentations [18]. It remains to be seen whether these types of models can be reconciled with our current findings.

## Model predictions

Our model makes several predictions which can be tested experimentally. First, the timescale of the short-term memory is intimately linked with the timescale of short-term depression (i.e. the recovery time constant parameter $\tau_x$ in our model). Short-term depression operates on the order of milliseconds to seconds, and we predict that in tasks with delay periods which exceed this timescale, the short-term memory will be lost as the synaptic efficacies are released from depression and return to baseline. We note that prior work [38] has suggested that thalamo-cortical synaptic depression may give rise to many of the suppression phenomena observed in visual cortex. This model also predicts that suppression can operate on the order of milliseconds and recovery from cross-orientation suppression is dependent on the orientation and duration of the flashed stimuli and mask.

Changing the length of the delay period during the task will allow us to estimate the timescale of synaptic depression and determine whether this is a fixed intrinsic property or can be adapted to the task at hand. Our model also predicts that interrupting the presentation of one or more of the repeated stimuli prior to the change image will also have an impact on behavioral performance. The effect of this manipulation will be to allow the synapses to recover, resulting in an inability to distinguish the repeat image from the change image. As a result of this manipulation, we also predict a higher false alarm rate on "catch" trials, as the memory of the repeated image has been altered. However, when pushing the system to a condition in which adaptation is no longer a useful cue, we believe that the animals will employ a different strategy, potentially using persistent neural activity to solve the task.

## Scope and limitations of the model

In the current study, our model only considers the responses of excitatory neurons, but understanding the contributions of different inhibitory cell types and the emergence of temporal expectation signals with experience remains an exciting area of future study [16]. We showed that short-term synaptic depression was sufficient to solve the change detection task. However, we did not consider the role of short-term synaptic facilitation, which may also play an important role in memory maintenance. On other memory tasks, the interaction of short-term facilitation and depression may be needed to account for more complex neural dynamics and behavior. Although our results point to an adaptation mechanism, we have not conclusively determined whether the exact mechanism is due to short-term synaptic depression or intrinsic cellular mechanisms (e.g. firing rate adaptation). Future work may be able to distinguish these two different mechanisms.

We used a pretrained convolutional neural network as a model of the mouse visual cortex to transform images into low-dimensional feature embeddings. Previous work has shown that

convolutional recurrent neural networks may be a better model of the primate visual system [39]. Similarly, short-term synaptic plasticity could be incorporated into the feedforward connections of the convolutional neural network. Understanding the role of short-term plasticity and recurrent connections in the feature pre-processing stage are interesting directions for future research.

Our current models emphasize the dichotomy between persistent neural activity and short-term synaptic plasticity mechanisms, which is unlikely to be true in the brain. We have also trained models with both mechanisms (STPRNN, S7 Fig). While we show a large improvement in the speed of learning for STPRNN, it should be noted that STPRNN has more parameters than either STPNet or RNN. Interestingly, despite having access to persistent neural activity, STPRNN produces behavioral patterns (as quantified by the matrix symmetry value) which are indistinguishable from the STPNet model and the experimental data, suggesting input adaptation plays a central role in this task. This could be due to the simplicity of the task, in which short-term memory only needs to be maintained and compared to the image presented, but does not need to be manipulated. This aligns with previous work [28] showing the need for maintenance of short-term memories in persistent activity only if the memories need to be manipulated.

The scope of the current work is constrained by comparison to biological data. However, we believe our work also has the potential to inform new model architectures in the field of artificial intelligence, where the neural networks currently used employ static synapses that have no history or temporal dependence. We believe that incorporating multiple dynamical variables may allow these models to be more robust and have the capability to store short-term memories.

## Supporting information

**S1 Fig. Schematic of how the change modulation index is computed.** For a given neuron or unit, A is the mean pre-change image response and B is the mean change image response. The change modulation index (CMI) is computed as $\frac{B-A}{B+A}$, or the difference between the change and pre-change image responses, divided by their sum (see Methods). A change modulation index greater than zero is indicative of adaptation (top right), while a change modulation index less than zero is indicative of facilitation (bottom right).
(EPS)

**S2 Fig. Decoder accuracy as a function of number of neurons.** Decoder accuracy for both image change (black) and image identity (red) increases with the number of neurons included in the analysis, saturating around 100 neurons. Neurons were added in increments of 10, randomly sampled using 10 cross-validated samples per experimental session. For each number of neurons, only sessions which had enough neurons were used in the analysis. Decoder accuracy is averaged across sessions and samples. Error bars show 95% confidence intervals on the mean.
(EPS)

**S3 Fig. Comparison of model hidden unit output weights and change modulation indices.** (A) Scatter plot of output weights and change modulation indices for the hidden units in the STPNet model ($N = 10$ models). Two clusters emerge, with one showing positive change modulation indices and output weights (corresponding to adaptation) and the other showing negative change modulation indices and output weights (corresponding to facilitation). A strong positive correlation ($r = 0.98$) is evident as for neural data from the experiments, suggesting that units which adapt are informative for change detection. (B) Same convention as in (A),

for the RNN model. There is also a strong positive correlation ($r = 0.97$).
(EPS)

**S4 Fig. Fraction of variance explained as a function of the number of principal components for the experiments (red), STPNet models (dark blue) and RNN models (light blue).** The results are averages over $N = 13$ experiments and $N = 10$ models for each type.
(EPS)

**S5 Fig. Low-dimensional analyses of population activity for an example experiment and trained examples of the two computational models.** (A) LDA analysis showing projection of population activity onto the first two LDA dimensions. Colors label the identity of the shown image and are the same for panels E and I below. The different images can be well separated in this low-dimensional space. (B) The same projection as in panel (A), with the color now showing the number of image repeats (more blue hue indicates higher repeats). Note the difference in number of repeats when compared to panels F and J below. The experiments show a "collapse" toward the origin of the low-dimensional space with increasing number of repeats. (C) Quantification of the population-level adaptation effect as a function of number of repeats. The distance from the origin of the low-dimensional space is plotted using either using the Euclidean distance (blue) or the absolute value of the first LDA dimension (orange). The experiment shows decreasing distances with repeats, and the fact that the first LDA dimension tracks the Euclidean distance indicates a low-dimensional effect. (D) Same convention as in panel (C), now showing the measured distances over all experimental sessions ($N = 13$). The mean and 95% confidence intervals are shown for each distance plot. (E-H). Same as (A-D), but for the STPNet model, which shows similar trends as for the experiments. (I-L) Same as (A-D), but for the RNN model. The RNN model shows differences with the experiments and STPNet model, mainly a constant distance as a function of image repeats.
(EPS)

**S6 Fig. Low-dimensional analyses of population activity for an example experiment and trained examples of the two computational models.** (A) Isomap analysis showing projection of population activity onto the first two isomap dimensions. Colors label the identity of the shown image and are the same for panels E and I below. The different images can be well separated in this low-dimensional space. (B) The same projection as in panel (A), with the color now showing the number of image repeats (more blue hue indicates higher repeats). Note the difference in number of repeats when compared to panels F and J below. The experiments show a "collapse" toward the origin of the low-dimensional space with increasing number of repeats. (C) Quantification of the population-level adaptation effect as a function of number of repeats. The distance from the origin of the low-dimensional space is plotted using either using the Euclidean distance (blue) or the absolute value of the first isomap dimension (orange). The experiment shows decreasing distances with repeats, and the fact that the first isomap dimension tracks the Euclidean distance indicates a low-dimensional effect. (D) Same convention as in panel (C), now showing the measured distances over all experimental sessions ($N = 13$). The mean and 95% confidence intervals are shown for each distance plot. (E-H). Same as (A-D), but for the STPNet model, which shows similar trends as for the experiments. (I-L) Same as (A-D), but for the RNN model. The RNN model shows differences with the experiments and STPNet model, mainly a constant distance as a function of image repeats.
(EPS)

**S7 Fig. Comparison of models with short-term synaptic depression and recurrent connections (STPRNN) to the RNN and STPNet models used in the paper.** (A) Average training trajectories of the different models (dark solid lines show means, light shaded regions show

standard deviations). The STPRNN model (red) trains much faster than the RNN (blue) and STPNet (green) models. (B) Quantification of the mean number of epochs required for each model to reach the dprime stopping criterion ($N = 10$ different random seeds). The color convention is the same as in panel (A). (C) Quantification of the mean $L_2$ weight norm for different connection types in the STPRNN model. The model learns larger magnitude weights on the depressed inputs (right) relative to the non-depressed inputs (center) and the recurrent weights (left). Note that during training, a small $L_2$ penalty was placed on the activations of hidden units. (D) Average response probability matrices which show the probability of responding to each of the possible 64 image transitions during the task (go trials off the diagonal, catch trials along the diagonal). The STPRNN (left) and STPNet (right) models show an asymmetry in the detectability of images, but not the RNN model (center). (E) Quantification of response probability matrix asymmetry. Matrix symmetry is bounded between –1 and 1, with negative values indicating asymmetry and positive values indicating symmetry. The STPRNN (left) and the STPNet (right) models show a small degree of matrix asymmetry, while the RNN model (center) shows high matrix symmetry. Error bars indicate 95% confidence intervals on the mean. (EPS)

**S8 Fig. Effect of varying $\tau_x$ value or the delay period on STPNet training performance.** (A) Effect of different $\tau_x$ values (ms), with the delay period held fixed at 250 ms as in the paper. (B) Effect of different delay periods (ms), with $\tau_x$ held fixed at 1.5 seconds as in the paper. Average training trajectories over $N = 10$ different model initializations are shown. (EPS)

**S9 Fig. Example recurrent weight distribution from a trained RNN model.** There is a total of 256 weights in the histogram (16x16 weights with a 16-dimension hidden layer). The trained models tend to show both positive and negative weights centered around zero. Weights are initialized from a uniform distribution in [-0.25, 0.25]. (EPS)

## Acknowledgments

We wish to thank the Allen Institute for Brain Science founder, Paul G. Allen, for his vision, encouragement, and support. We would also like to thank Matthew Valley, Corbett Bennett, and Alex Piet for helpful discussions.

## Author Contributions

**Conceptualization:** Brian Hu, Stefan Mihalas.

**Data curation:** Marina E. Garrett, Peter A. Groblewski, Douglas R. Ollerenshaw, Kate Roll, Sahar Manavi.

**Formal analysis:** Brian Hu.

**Funding acquisition:** Christof Koch.

**Investigation:** Brian Hu.

**Methodology:** Brian Hu, Jiaqi Shang.

**Resources:** Marina E. Garrett, Peter A. Groblewski, Douglas R. Ollerenshaw, Kate Roll, Sahar Manavi.

**Software:** Brian Hu, Marina E. Garrett, Douglas R. Ollerenshaw, Jiaqi Shang.

**Supervision:** Christof Koch, Shawn R. Olsen, Stefan Mihalas.

**Validation:** Brian Hu.

**Visualization:** Brian Hu, Marina E. Garrett.

**Writing – original draft:** Brian Hu, Stefan Mihalas.

**Writing – review & editing:** Brian Hu, Marina E. Garrett, Peter A. Groblewski, Douglas R. Ollerenshaw, Jiaqi Shang, Kate Roll, Sahar Manavi, Christof Koch, Shawn R. Olsen, Stefan Mihalas.

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
