## [Decision Letter · Decision Letter 0]

30 Dec 2020

Dear Dr. Mihalas,

Thank you very much for submitting your manuscript "Adaptation supports short-term memory in a visual change detection task" for consideration at PLOS Computational Biology.

As with all papers reviewed by the journal, your manuscript was reviewed by members of the editorial board and by several independent reviewers. In light of the reviews (below this email), we would like to invite the resubmission of a significantly-revised version that takes into account the reviewers' comments.

We cannot make any decision about publication until we have seen the revised manuscript and your response to the reviewers' comments. Your revised manuscript is also likely to be sent to reviewers for further evaluation.

Sincerely,

Boris S. Gutkin

Associate Editor

PLOS Computational Biology

Wolfgang Einhäuser

Deputy Editor

PLOS Computational Biology

Reviewer's Responses to Questions

**Comments to the Authors:**

Reviewer #1: In this paper the authors compare two neural network models with neural data obtained from a visual cortical area as mice learn to solve a task. In particular, the authors implement two models which utilise different mechanisms underlying short-term memory: the STPNet using short-term synaptic depression and an RNN model using persistent activity through recurrent weights. These models were compared with neural and behavioural data taken from a mouse performing a visual change detection task. The results seem to suggest that the STPNet model better captures the neuronal data. This combination of data and model is novel and is of potential interest to not just computational neuroscience but also the wider neuroscience community. The paper is well written, but there is a lack of clarity in several places. In addition, at the present there are substantial issues with the paper that should be addressed.

Major points

1. The authors pretrained a CNN which was used to generate low-dimensional features to both models considered. CNNs are often interpreted as models of the visual cortex (e.g. work from the Dicarlo lab). This means that the connections within the CNN model should themselves have STP. Similarly, a recurrent CNN (e.g. Nayebi et al. NeurIPS 2018) could in principle be studied. These two models could then be compared with one another. One option would be to adapt the CIFAR dataset to directly train the task that mice were trained on, potentially augmenting the dataset used for animal training with a larger dataset. We understand that this might complicate things, but it should be at least discussed in detail and these two components of the model better explained (see also major point 3 below).

2. In the STPnet the STD parameters are fixed (apart from W which are learnt via backprop). Whereas in the RNN model all parameters are learnt via backprop. This makes the comparison somewhat unfair as the dynamic component of the RNN (i.e. the recurrent weights) are modified through backprop but the equivalent component in the STPnet is not. What would happen if the recurrent weights in the RNN are fixed to values equivalent to the STD dynamics (e.g. w_rec ~ 0.5)? Conversely what happens if you also train the STD parameters using backprop? There is wide evidence to support that STP is plastic in itself (see for example a model that captures this interaction Costa et al. 2015 eLife). In general, the current RNN model does not seem to have any decay, which is odd and not consistent with recurrency in the brain as the effective recurrent weights tend to be relatively low (e.g. w_rec ~ 0.1), which should give you a decay. Adding this extra model experiments would help to clarify these points.

3. Figure 2 is critical to understand the two proposed models. However, in its current form it is hard to understand and contrast the two models. For example, the full architecture could be presented in one panel highlighting that the bottom part of the model is a CNN that is pretrained on image recognition and then on top the two models further trained to do the task considered in this paper. In addition, it would be instructive to also show the output for the RNN in C-E. Also, LTP is usually used to refer to Long-term potentiation, so it would clarify things if LTSP (Long-term syn. plast.) was used instead. y-axis in D and E is missing units.

4. To make it possible to understand how the models are linked to the data, the models and how they were trained need to be (briefly) explained in the Results section. This should be done before describing the model results in the "Asymmetry in the detectability" section. Also, the main text never seems to go into the detail of Fig. 2, please add this to the main text.

5. There is a lack of statistical tests throughout to support the conclusions. For example, in Fig. 3 B,C and E. But also in several other places (Fig. 6C).

6. There are many important details lacking in the methods that would be essential to be able to reproduce and fully understand the work. For example the exact model used for the RNN is not given. See also the various minor points below.

Minor points

1. Title is perhaps not a good reflection of the study as adaptation can mean many different things. For example "Synaptic adaptation.." would be more accurate.

2. L21: It has been known for some time now that most synapses in the visual cortex are depressing (e.g. work by Henry Markram in the 90s and many others, see also Buchanan et al. 2012 Neuron for excitatory synapses onto other cell types).

3. This work is directly relevant to the present study and should be discussed: https://www.ncbi.nlm.nih.gov/pmc/articles/PMC6757815/

4. Fig1: The caption should have more details. For example in B is not clear what are the two images exactly showing. Can you please also increase the resolution and size of the figures in B so that they are easier to read.

5. Methods: The actual equations of the firing rate approximation of the STD model should be given for clarity.

6. Methods: A L2 penalty was used to encourage sparsity. Why is this needed?

7. Fig. 5: It would be interesting to plot the variance explained vs number of components for all the 3 cases.

8. Fig. 4: A quantitative comparison between model and data modulation indexes, and the respective summary plot would be useful here.

9. Please clarify any pre-processing performed on the input images and if the CNN was fine-tuned on the images that were used during the (changing) task training. This is important to state since the CNN was pretrained on CIFAR-10, which is different from the input images used later on.

10. It would be interesting to have a high-level explanation for the importance of asymmetries in the responses to the changed images. It is not immediately obvious what the significance is of the fact that STPNet and the behavioural data show this asymmetry in the response and the RNN does not (apart from the quantitative comparison).

11. It would be interesting to show model experiments that vary the time between the presentation of each stimulus to compare the decay properties of the neural data vs the STPNet (this point is touched on in the "Model Predictions" section).

12. Related to the previous point and major point 2. It would be important to see how the models behave for different tau depression (STPnet) and recurrent weights (RNN).

13. Its not clear exactly how many parameters the different models use, please add these to the text or ideally a table with this and the full hyperparamaters.

14. The change modulation index is not trivial to understand, adding a supp. figure that explains the index would be great.

15. (in abstract): "Unlike the RNN model, STPNet also produces a similar pattern of behavior". Remove 'also'.

16. Fig 2d caption: "Input-dependent changes in synaptic efficacy..." we assume that this refers to the STP model but this could be better specified.

17. Section "Low-dimensional analyses of neural and model activity" talks about "Euclidean distance of the full population activity", but corresponding Fig. 5c caption says "The distance

from the origin of the low-dimensional space is plotted using either using the Euclidean distance..". This seems to imply euclidean distance of lower-dimensional space (not full activity as implied in main text). Please clarify.

18. In the "Model predictions" section it is mentioned that "the timescale of the short-term memory is intimately linked with the timescale of short-term depression" to solve the task. This is not necessarily a novel prediction as it is a directly follow up from other studies https://www.ncbi.nlm.nih.gov/pmc/articles/PMC6757815/. Please clarify.

19. Methods: Clarify that the 10 random seeds correspond to 10 different model initialisations. What init. was used?

20. Fig. 3: Although the exact performance of the CNN is not relevant, it is relevant to show that it was trained to get reasonable accuracy on the CIFAR task to this end a standard accuracy measure should also be shown.

Reviewer #2: The authors present two competing models for the behavior and neural activity of mice tested in a visual change detection task. One model is a recurrent neural network while the other is a network with short-term depression between the input units and the hidden layer, with not recurrent connections in the latter. The idea is to test which mechanism best captures the data and is therefore a better candidate for this particular form of short-term memory. Both networks have a small number of neurons, presumably because the task is simple to solve. The main claim of the paper is that "while both networks are able to learn the task, the STPNet model contains units whose activity are more similar to the in vivo data and produces errors which are more similar to the mice."

The authors have compared other metrics (based on dimensionality reductions, change adaptation indices in single neurons, and other quantities), and they all seem to point to the same results. They also tested their results with a network that "interpolates" between the RNN and the STPnet, i.e., it becomes one or the other net in two opposite special cases. The latter network works best, but the main signature of STP is still there and accounts for properties of the neural data that the RNN alone cannot capture.

Although this study cannot rule out the role of persistent activity (as subserved by the RNN) in short-term memory in general, their results are quite convincing that adaptation plays an important role in explaining this data. They also admit that a crucial test of their prediction would be to test the mice in a version of the task with variable inter-trial interval, since the model with STP makes clear predictions based on the time constant of recovery from depression. It is possible that both mechanisms are used, perhaps with STP being more relevant with constant (and short) intertribal intervals (as here), while persistent activity being more relevant in more difficult tasks with highly variable delay. It seems clear that, in the latter case, having a mechanism that can retain the representation of the last image would be more robust than one relying on a restricting range of adaptation processes (although adaptation on multiple and heterogeneous time scales has also been observed, and having neurons with very diverse time constants perhaps could do the job).

In conclusion, I believe the authors make a fair claim, at least in relation to this particular change detection experiment. Their methodical procedures seem appropriate to derive their conclusions.

- Minor points:

l. 146: "(compare Figure 5G,L). These findings held over the ensemble of 146 models that were tested (Figure 5H,M)". Panels G,K and H,L instead?

Reviewer #3: The authors investigate, by means of experiments and computational modeling, neuronal dynamics during a visual change detection task which requires short-term memory. They compare against experimental data, behavioral and neuronal, two models: one supporting memory storage via persistent activity (RNN), the other via short-term synaptic dynamics in the form of short-term depression, in the absence of persistent activity (STPNet). It turns out that STPNet explains better than RNN both the behavioral pattern observed (asymmetry in the detections) and the neuronal dynamics (presence of significant repeat effects in the neuronal response), to some extent.

The results presented appear correct (as far as I can say), novel to some extent, and interesting. The paper is clearly written and I found the figures well chosen and informative.

I have a few, essentially minor, comments.

There is a fairly large literature (experiments and modeling) on so-called match effects in non-human primates that seems, prima facie, to be relevant in the present context. For instance, the idea that some form of neuronal or synaptic “fatigue” could be responsible for repetition suppression (as observed and modeled here) is certainly not novel. I think the authors should shortly discuss the relevance (or not) of these studies for their results.

Partly related to the above, Tartaglia et al. (2014) have argued that, at least in non-human primates, repetition suppression is always accompanied by match enhancement and persistent activity. At least for the match enhancement, this seems to be true also here. By looking at Fig. 4C, there is a small, but significant I would guess, fraction of neurons that have a negative change modulation index (CMI). Interestingly (if I read Fig. 4D correctly), STPNet is unable to produce negative a CMIs, while RNN produces negative (though very small compared to experiment) CMIs. Please comment on these points.

Is there any persistent activity in the experimental recordings?

**Have all data underlying the figures and results presented in the manuscript been provided?**

Reviewer #1: **No: **The authors state that the data is available in the metadata but I failed to find any link or pointers to this in the actual paper.

Reviewer #2: None

Reviewer #3: Yes

PLOS authors have the option to publish the peer review history of their article (what does this mean?). If published, this will include your full peer review and any attached files.

Reviewer #1: No

Reviewer #2: No

Reviewer #3: No
---

## [Decision Letter · Decision Letter 1]

3 Jul 2021

Dear Dr. Mihalas,

We are pleased to inform you that your manuscript 'Adaptation supports short-term memory in a visual change detection task' has been provisionally accepted for publication in PLOS Computational Biology.

Best regards,

Boris S. Gutkin

Associate Editor

PLOS Computational Biology

Wolfgang Einhäuser

Deputy Editor

PLOS Computational Biology

Reviewer's Responses to Questions

**Comments to the Authors:**

Reviewer #1: The authors have done a relatively good job at addressing the points raised. There are still open issues which do not permit to have a good comparison between the models considered, but as discussed by the authors these could be part of a future study.

Reviewer #3: The authors have satisfactorily addressed all issues I raised in my previous report.

**Have the authors made all data and (if applicable) computational code underlying the findings in their manuscript fully available?**

Reviewer #1: Yes

Reviewer #3: Yes

PLOS authors have the option to publish the peer review history of their article (what does this mean?). If published, this will include your full peer review and any attached files.

Reviewer #1: No

Reviewer #3: No

---

## [Editor Report · Acceptance letter]

22 Aug 2021

PCOMPBIOL-D-20-01684R1 

Adaptation supports short-term memory in a visual change detection task

Dear Dr Mihalas,

I am pleased to inform you that your manuscript has been formally accepted for publication in PLOS Computational Biology. Your manuscript is now with our production department and you will be notified of the publication date in due course.

With kind regards,

Zita Barta
